



# Evaluation of the impact of wood combustion on benzo(a)pyrene concentrations, using ambient air measurements and dispersion modelling in Helsinki, Finland

Heidi Hellén[1], Leena Kangas[1], Anu Kousa[2], Mika Vestenius[1], Kimmo Teinilä[1], Ari Karppinen[1], Jaakko Kukkonen[1] and Jarkko V. Niemi[2,3]

[1] Finnish Meteorological Institute, P.O. Box 503, FI-00101 Helsinki, Finland
[2] Helsinki Region Environmental Services Authority, P.O. Box 100, FI-00066 HSY, Helsinki, Finland
[3] Department Environmental Sciences, University of Helsinki, P.O. Box 65, FI-00014 University of Helsinki, Finland

*Correspondence to*: Heidi Hellén (heidi.hellen@fmi.fi)

**Abstract.** Even though emission inventories indicate that wood combustion is a major source of PAHs, estimating its impacts on PAH concentration in ambient air is challenging. In this study effect of local small-scale wood combustion on the
benzo(a)pyrene (BaP) concentrations in ambient air in the Helsinki metropolitan area in Finland was evaluated, using ambient air measurements, emission estimates and dispersion modelling. Measurements were conducted at 12 different locations during a period from 2007 to 2015. The spatial distributions of annual average benzo(a)pyrene concentrations originated from wood combustion were predicted for four years: 2008, 2011, 2013 and 2014. According to both the measurements and the dispersion modelling, the European Union target value for the annual average BaP concentrations (1 ng m$^{-3}$) was clearly exceeded in part
of the suburban detached house areas. However, over most of the other urban areas, including the centre of Helsinki, the concentrations were below the target value. The measured BaP concentrations were highly correlated with the measured levoglucosan concentrations at suburban detached house areas. In street canyons, the measured concentrations of BaP were at the same level as urban background, being clearly lower than those in suburban detached house areas. The predicted annual average concentrations agreed fairly well with the measured concentrations. Both measurements and modelling clearly
indicated that wood combustion was the main local source of ambient air BaP in the Helsinki metropolitan area.

Keywords:  PAHs, PM$_{2.5}$, air pollution, air quality, levoglucosan





## 1 Introduction

Wood is widely used as a fuel for residential heating in many countries. However, residential wood combustion can have a significant effect on air quality by emitting substantial quantities of fine particles ($PM_{2.5}$) and other pollutants (Simoneit et al., 2002). Polycyclic aromatic hydrocarbons (PAHs) are known to be carcinogenic constituents of fine particles (Ravindra et al.,

2008). They are produced due to incomplete combustion of biomass, coal, oil, and gasoline and diesel fuels. Benzo(a)pyrene (BaP) has been regarded as a marker of both the total and carcinogenic PAHs (EC, 2004). Even though concentrations of many pollutants have decreased in European Union area during last 20 years, any significant trend for most PAHs was not found in background air of Southern Sweden and Northern Finland (Anttila et al. 2016). This could be related to the ongoing and not decreasing sources.

European Commission has set a target value of 1 ng $m^{-3}$ for the annual average concentration of BaP in ambient air (e.g., European Environment Agency, 2015). The ambient air concentrations of PAHs and BaP have been of concern in Europe, since the concentration levels have been relatively high with respect to the target values (European Environment Agency, 2013, Guerreiro et al., 2015). During 2011-2013, 25-29% of the urban population in the EU was found to be exposed to

concentrations that were above the above mentioned target value of BaP. If we will consider reference level of World Health Organization (WHO), i.e., 0.12 ng $m^{-3}$, 85-91% of the urban populations in the EU were exposed to BaP values that were higher than the reference level (European Environment Agency, 2015). The reference level was estimated assuming WHO unit risk for lung cancer for PAH mixtures, and acceptable risk of additional lifetime risk of approximately 1 x $10^{-6}$.

There is evidence that residential wood combustion is a significant source of airborne PAHs at many locations. Butt et al. (2016) modelled the impact of residential combustion emissions on atmospheric aerosol in 2000, using a global aerosol microphysics model. The largest contributions of residential emissions to annual surface mean $PM_{2.5}$ concentrations were found to occur in East Asia, South Asia, and Eastern Europe. Mortality due to residential emissions was found to be greatest in Asia, with China and India accounting for 50% of simulated global excess mortality.

Guerreiro et al. (2015) evaluated on a European level the main emission sources of BaP, the concentration levels, population exposure and potential health impacts. They used a mapping methodology, which combines monitoring data with modelling data and other supplementary data. They found the ambient air concentrations of BaP to be substantially high especially in central and central Eastern Europe, but also in some other European regions. The highest concentrations were interpreted to

be mostly due to emissions from the domestic combustion of coal and wood.

Silibello et al. (2012) modelled the BaP concentrations in Italy. The analysis revealed a significant influence of national sources on BaP concentrations; the most important emission sector was non–industrial combustion using wood burning devices. In





Northern Italy ambient air measurements and cluster analysis indicated that wood combustion was the main source of BaP at all other sites, except for the city of Milan (Gianelle et al., 2013; Belis et al., 2011). In Augsburg, Germany, at a site representing a typical inner city residential location, the contribution of wood combustion to measured PAH levels was estimated to be as high as 80–95% (Schnelle-Kreis et al., 2007). In the UK, positive matrix factorization results indicated that wood combustion

had an important role on the PAH concentrations in urban air, even though traffic and coal combustion were found to be the main sources (Jang et al., 2013).

Wood burning for domestic heating is a major emission source of primary $PM_{2.5}$ in many Nordic cities and its role may become even more pronounced in the future, if wood consumption continues to increase (Savolahti et al., 2016). Denby et al. (2010)

evaluated the source contribution of wood burning at measurement sites in the Norwegian cities of Oslo and Lycksele to be around 25% for the $PM_{2.5}$ concentrations. In the Helsinki metropolitan area, the $PM_{2.5}$ emissions originated from residential wood burning were found to be 39% of the total annual emissions of all combustion sources (Kaski et al., 2016a,b). Residential wood burning in the Helsinki metropolitan area is spatially concentrated on suburban detached house areas, in which substantial amounts of PM can be emitted from various wood combustion devices, such as, e.g., heat storing masonry heaters

and sauna stoves.

For PAHs wood combustion may be even more significant source than for $PM_{2.5}$. In Denmark, residential wood combustion has been estimated to account for about 90% of the Danish emissions of BaP (Glasius et al., 2008). In Central Finland, the measured PAH concentrations have been found to be several times lower in background air than in a small residential area

including 164 detached houses, in which wood was used as a secondary energy source (Hellén et al., 2008).

However, the scientific information is scarce regarding the quantitative effects of residential wood combustion on the ambient air concentrations of PAHs. There are also very few studies regarding the spatio-temporal variation of such concentrations in urban air. This kind of research has previously been seriously hampered by problems in reliably estimating the spatial

distributions and temporal variation of the emissions originated from wood combustion.

Even though emission inventories indicate that wood combustion is a major source of PAHs (e.g., Shen et al., 2013), estimating wood combustion emissions is challenging, as these depend on numerous factors, such as the type and construction of a fireplace, operating procedures and the quality of the wood used (e.g., Ozgen et al., 2014; Tissari et al., 2007 and 2009;

Savolahti et al., 2016). The information that can be obtained based on simultaneous ambient air concentration measurements and dispersion modelling is therefore crucial for estimating the effects of wood combustion emissions on ambient concentration levels and the exposure of populations.





The main aim of this study is to quantitatively evaluate the impacts of wood burning on the concentrations of BaP in the Helsinki metropolitan area. We have conducted ambient air measurements during several years, compiled a new emission inventory and modelled atmospheric dispersion for four target years. The concentrations of levoglucosan, a source-specific tracer for biomass burning particles, were also measured and compared with the concurrently measured concentrations of BaP.

**2 Methods**

**2.1 Site descriptions and sampling periods**

The Helsinki metropolitan area (HMA) comprises four cities; Helsinki, Espoo, Vantaa and Kauniainen. The total population in the HMA is approximately 1.1 million, while the population of Helsinki is about 0.63 million inhabitants. The contributions
of the different emission source categories for the total combustion emissions of $PM_{2.5}$ in HMA in 2015 were 39% for small-scale wood combustion, 31% for energy production and other stationary sources, 28% for vehicular traffic and 2% for harbors, according to Kaski et al. (2016a,b).

The centre of Helsinki is located on a peninsula that is surrounded by the Baltic Sea; the main detached house areas are situated
to west, east and north from the city centre (Fig. 1). The annual mean temperature in Helsinki is 5.9 °C. However, the seasonal variation of the temperatures is substantial; monthly mean minimum and maximum is -4.7 °C in February and 17.8 °C in July, respectively. Heating in the HMA is mainly based on an extensive district heating system that has a minor impact on air quality. The reason for this is that the heat is mainly obtained from plants burning fossil fuels; most of these plants have very high stacks. However, fireplaces and sauna stoves are commonly used in suburban detached houses.

The measurement sites used in this study (Table 1 and Fig. 1) in the HMA were situated as follows: eight sites were in detached house areas (2008–2015), one in urban background (2007–2015), and three within street canyons (2007, 2010 and 2015). The monitoring height at all stations was approximately 4 m. The urban background station of Kallio is situated in a sports field in the city centre; its distance from the closest street with a traffic volume of 6300 vehicles day$^{-1}$ is approximately 80 m. The
stations in detached house areas, numbers 1-8 (Vartiokylä, Itä-Hakkila, Päiväkumpu, Kattilalaakso, Kauniainen, Tapanila, Ruskeasanta and Lintuvaara) were situated in suburban areas with relatively lower traffic volumes. Street canyon stations (Unioninkatu, Töölöntulli and Mäkelänkatu) were on busy streets with high traffic volumes and $NO_x$ concentrations (Table 1). Average traffic density at Unioninkatu was 12 800 vehicles weekday$^{-1}$ (7% heavy traffic), at Töölöntulli 44 000 vehicles weekday$^{-1}$ (10% heavy traffic) and at Mäkelänkatu 28 000 vehicles weekday$^{-1}$ (9% heavy traffic).


Data from the stations in the HMA was compared with data from rural and remote stations. Rural station 1 of Virolahti (60°31'N, 27°40'E, 5 m a.s.l) in Eastern Finland is located at a distance of 160 km from Helsinki in a rural district on the coast of the Gulf of Finland (Vestenius et al. 2011). Rural station 2 that is situated at a distance of 210 km north from Helsinki




in Hyytiälä (61º51'N, 24º17'E, 181 m a.s.l) is a boreal forest site that is part of the SMEAR network, SMEAR II (Station for Measuring Ecosystem-Atmosphere Relationships) in southern Finland (Hari and Kulmala, 2005). Remote station of Pallas is situated in Matorova, Northern Finland (68°00'N, 24°14'E, 306 m .a.s.l.), at the distance of 900 km from Helsinki. Pallas is in the subarctic region at the northernmost limit of the northern boreal forest zone (Hatakka et al. 2003).

## 2.2 Measurement methods

The daily $PM_{10}$ samples were collected on polytetrafluoroethylene (PTFE) filters (FluoroporeMembraneFilters, 3.0 μm, Ø 47mm, Merck Millipore Company, Germany) every 2-4th day by MicroPNS –low volume samplers. The flow rate used was

38 l min$^{-1}$ and the average total collected volume for 24-hour samples was 55 m$^3$. For the analysis samples were usually pooled together as monthly samples, soxhlet extracted with dichloromethane, dried with sodium sulfate, concentrated to 1 ml and cleaned using Florisil solid phase extraction (SPE) catridges. After that the concentrations of BaP were analysed by gas chromatograph-mass spectrometers (GC-MSs, Agilent 6890N and 5973).In the analysis of BaP, the ISO 12884 (2000) and EN 15549 (2008) standards were followed. Measurement uncertainty at the target value (1 ng m$^{-3}$) was estimated to be 11%. The

method has been previously described in detail by Vestenius et al. (2011).

The levoglucosan concentrations were determined from daily $PM_{10}$ samples from urban background and detached house areas in 2012 and from monthly means in 2011. Samples were collected on PTFE filters, extracted with 5 ml of deionized water with internal standard and analyzed with high performance anion exchange chromatography-mass spectrometry as described

by Saarnio et al. (2012).

## 2.3 Evaluation of BaP emissions from wood combustion

The emissions of BaP originated from small-scale wood combustion were evaluated in the HMA, including the spatial and

temporal variation in emissions. We did not evaluate the emissions of BaP originated from local vehicular traffic, or other potential local sources. Vehicular traffic has been shown to have a minor influence on the total emissions of BaP on European scale (Guerreiro et al. 2015); also based on our ambient air measurements at traffic sites (see Section 3.1.) emissions of BaP from traffic are low in the HMA. In addition to this there are no potential industrial or other local sources of BaP in Helsinki (Soares et al. 2014).


A novel wood combustion emission inventory was compiled for the HMA in 2014 for the following pollutants: particles ($PM_1$, $PM_{2.5}$ and $PM_{10}$), nitrogen oxides (NOx), non-methane volatile organic compounds (NMVOC), carbon monoxide (CO), black carbon (BC) and benzo(a)pyrene (BaP). For a more detailed description of this inventory, the reader is referred to the report




by Kaski et al. (2016a). The BaP emissions originating from wood combustion depend on numerous factors, such as the type and construction of a fireplace, operating procedures and habits, and the quality, processing and storage of the wood used.

The amount of wood combusted, and the procedures and habits were estimated by a questionnaire in the HMA (Kaski et al., 2016a). The aim of the questionnaire was to gather quantitative information on the amount of the wood combusted and the combustion characteristics, in order to evaluate its impacts on air quality. The questionnaire was sent to 2500 inhabitants of detached house areas. The response rate was 35%. A stratified sampling procedure was used to ensure the representativity of the replies. The stratification procedure included the following three parameters of the houses; spatial distribution in the HMA, construction year and primary heating method.  In Finland, residential wood combustion is not common as a primary heating method, but it is substantially more frequently used as a method for supplementary heating. According to official statistics by the Statistics Finland and the Helsinki Region Environmental Services Authority (HSY), the total number of detached and semidetached houses in the HMA in 2014 was about 69 000. Most (52%) of the detached and semidetached houses were primarily heated with electricity. The shares of other primary heating methods were as follows; 22% oil or gas combustion, 18% district heating, 4% geothermal heating, 2% wood combustion and 3% unknown heating method (Statistics of Finland).

According to the questionnaire (Kaski et al., 2016a), wood combustion was used in approximately 90% of the detached and semidetached houses in the HMA. However, wood combustion was seldom used as a primary heating method in the detached and semidetached houses (only in approximately 2% of the houses); it was much more common as a supplementary heating method, and as fuel in the stoves of saunas. The annual average amount of wood burned per house was 1.52 solid cubic meters. Most of the wood was used in heat-storing masonry heaters (0.72 solid-m$^3$/house) and sauna stoves (0.31 solid-m$^3$/house) and only minor amount was burned in boilers (0.09 solid-m$^3$/house). In addition, there are also several other fireplace types to use the wood as a fuel (0.40 solid-m$^3$/house; e.g. open fireplaces, ovens, stoves).

The BaP emissions from wood combustions were calculated using the following emission factors: 809, 68 and 102 µg MJ$^{-1}$, for sauna stoves, boilers and other fireplace types (e.g. heat-storing masonry heaters, open fireplaces, ovens, stoves), respectively (Tissari et al., 2007; Todorović et al., 2007; Hytönen et al., 2009; Lamberg et al., 2011). The total BaP emission from wood combustion was estimated to be 196 kg in the HMA in 2014. The shares of BaP emissions were 2% for primary heating boilers, 67% for sauna stoves and 31% for other fireplace types. The pollutant emissions from sauna stoves are very high since their combustion conditions are usually very poor compared to other fireplace types (e.g. Savolahti et al., 2016 and references therein). Unfortunately, quite limited number of studies provide BaP emission factors for typical fireplace types used in Finland. Therefore, it would be very important to perform new combustion experiment studies to achieve more robust knowledge on BaP emission factors for sauna stoves and various other fireplace types as well as for different burning conditions.





The total amounts of wood burned and its allocation to different fireplace types is dependent on the primary heating method of a house (Kaski et al., 2016a). Spatial distribution of emissions for dispersion modelling was calculated using the following annual BaP emissions estimates for houses using different primary heating methods; 2.5, 3.7, 2.0, 4.1, 3.9 and 3.1 g/house for electricity, thermal, district, oil, wood and unknown heating methods, respectively (Kaski et al., 2016a). The previously
mentioned values contain the total emissions from the all fireplace types of a house, including sauna stoves. For the spatial allocation of emissions, the geographical location and primary heating method information of all 69 000 detached and semidetached houses of the HMA was available from the regional basic register (SePe and SeutuCD) provided by the HSY. The spatial distribution of houses and BaP emissions are shown in Fig. 1.

The temporal patterns (month, weekday, time of day) of emissions for three different fireplace categories (sauna stoves, boilers and other fireplaces) were estimated based on the information from the questionnaires (Kaski et al., 2016a; Gröndahl et al., 2011). Unfortunately, we did not get enough input information from the questionnaires to estimate the influence meteorological variables, such as temperature, on the emissions on daily or interannual level.

**2.4 Atmospheric dispersion modelling**

The atmospheric dispersion of BaP emissions was evaluated with the Urban Dispersion Modelling system that has been developed at the Finnish Meteorological Institute (UDM-FMI). The system includes various local scale dispersion models and a meteorological pre-processor MPP-FMI (Karppinen et al., 1998 and 2000a). The dispersion modelling of UDM-FMI is based
on multiple source Gaussian plume equations for various stationary source categories (point, area, and volume sources). For the selected calculation grid, the system was used to compute an hourly time series of concentrations. The modelling system has been evaluated by Karppinen et al. (2000b).

Meteorological input data needed by the dispersion model was evaluated using meteorological pre-processing model MPP-
FMI, based on the energy budget method. The model utilises meteorological synoptic and sounding observations, and its output consists of hourly time series of relevant atmospheric turbulence parameters and atmospheric boundary layer height. We used a combination of synoptic observations from the stations of Helsinki-Vantaa (15 km north of the city center) and Helsinki-Harmaja (on an island 7 km south of the city center), and the sounding observations from Jokioinen (90 km northwest of Helsinki). The predicted meteorological parameters vary for each hour of the year, but for each hour, the same value is applied
for the whole spatial domain.

In this study, we have evaluated the dispersion of BaP originated from domestic wood combustion. Emissions were uniformly distributed in squares of size 100m x 100m, and the model was applied to calculate the dispersion originated from these area sources. The altitude of the releases for domestic wood combustion was assumed to be equal to 7.5 m, including the initial





plume rise. This altitude value is based on the average heights of the detached and semidetached houses and their chimneys within the study domain, and an estimated plume rise (Karvosenoja et al., 2010).

The dispersion was computed separately for three different emission source categories: sauna stoves, boilers, and other fireplaces. The diurnal, weekly, and monthly variations within the emission inventory were applied for each source category. In the dispersion modelling, BaP was treated as an inert gas, i.e., no chemical or physical transformation was assumed to take place within the urban time scales. We also did not allow for the dry or wet deposition of BaP. The concentrations were computed for the years 2008, 2011, 2013, and 2014 to a receptor grid with a horizontal grid spacing of 100m x 100m. The influence of terrain on the atmospheric dispersion is parameterised simply as a surface roughness.

### 3 Results and discussion

### 3.1 Measured concentrations

#### 3.1.1 Annual and seasonal variation of BaP concentrations at different stations

The measured annual means of BaP concentrations at different stations in Finland have been presented in Fig. 2 and Table 1.
The highest concentrations were measured in suburban residential areas and the lowest ones at the regional background and remote sites. The BaP concentrations at the street canyon sites (SCs) were quite low and at the same level as at the urban background site (UB). The EU target value for BaP (1 ng m$^{-3}$) was exceeded at two of the residential sites, and the measured concentrations were in the vicinity of the target value at a few other residential sites. However, at all the other sites, the concentrations were well below the target value. The WHO reference level for BaP (0.12 ng m$^{-3}$) was exceeded at all urban
and suburban sites every year and at rural background sites during some years.

The variation of the annual average BaP concentrations can be compared with the corresponding variation for the PM$_{2.5}$ concentrations (Fig. 3). In case of PM$_{2.5}$, the highest concentration was observed at the busiest street canyon site (SC2). The WHO guideline value for PM$_{2.5}$ (10 µg m$^{-3}$) was exceeded both at the street canyon site (SC2) and at two of the suburban
residential sites (DH3 and DH7). However, PM$_{2.5}$ concentrations were below the WHO guideline value at most of the residential sites and at every background site.

These results indicate that local traffic has only a minor effect on the BaP concentrations, compared with the corresponding effect of small-scale combustion. The increase of the BaP concentrations caused by the regional and long-range transport is





noticeable. The BaP concentrations at one of the regional background sites (RB1) were approximately at the same level as those at the urban background and street canyon sites (Fig. 2). However, the mean BaP concentration at the remote site of Pallas in northern Finland was clearly lower.

The inter-annual variation of the BaP concentrations at the urban background site (UB) has been modest during 2007–2014. The corresponding variation has been higher at the residential site 1 (DH1) and at the rural background site 1 (RB1).

The BaP concentrations at all stations in Helsinki have a clear seasonal cycle, with the highest values in winter and the lowest ones in summer. At the suburban residential areas (DH1-8), the highest monthly mean values in winter were 1-4 ng m$^{-3}$,

depending on site and year. In summer the highest monthly values were usually below 1 ng m$^{-3}$. The monthly variation of BaP concentrations is illustrated in the next sections (see Figs. 4b and 5) and as supplementary information (Fig. S1). In case of PM$_{2.5}$, the seasonal variation is substantially smaller and different compared to that of BaP. Prevedouros et al. (2004) showed that at many European sites this seasonal trend of the BaP concentrations is mainly explained by the relatively lower emissions in summer; however, occasionally meteorology and air mass transport can change these patterns. Also reactions of BaP are

faster and lifetime shorter during summer (Keyte et al. 2013).

### 3.1.2 Correlation of the concentrations of BaP and levoglucosan

Levoglucosan has been shown to be a specific tracer compound for biomass burning, such as residential wood combustion and wild-land fires (Simoneit, 2002; Yttri et al., 2005; Saarikoski et al., 2008; Niemi et al., 2009; Saarnio et al., 2012). Even if levoglucosan may not be quantitative tracer due to its reactivity (Hennigan et al., 2010) and dependency on combustion conditions (Hedberg et al., 2006), its concentrations are still useful in tracking the biomass combustion emissions.

The correlation between the 24-h mean concentrations of BaP and levoglucosan in February 2012 in Helsinki was very high (correlation coefficient R$^2$=0.91, Fig. 4a). Also monthly mean concentrations in 2011 had a high correlation (R$^2$=0.82, Fig. 4b). If only winter months (Jan, Feb, Dec) are considered the correlation is even higher (R$^2$=0.88, N=12). Measured monthly means of BaP and levoglucosan are presented in supplements as figure S2. These high temporal correlations add more confidence to our finding that wood burning is the most important local source for BaP in this area. The above mentioned daily

and monthly correlations were high not only at the detached house areas, but also in urban background outside the detached house areas, indicating that the variations of the regional and long-range transport is responsible for part of those temporal correlations. Correlation coefficient squared (R$^2$) for 24-h means at urban background is 0.82, if one of the 13 parallel samples is removed as an outlier.




Average ratio of BaP and levoglucosan was 0.01. In biomass burning emissions average ratio has been 0.0011 (Belis et al. 2011). Our measurements were ambient air measurements and therefore differences in reactivity of the compounds in air may explain part of the difference, but also differences in used fuel and way of burning are expected to have significant effect on the ratio.

### 3.1.3 The effect of sauna stoves and other fireplaces on temporal variation

Sauna stoves heated with wood are known to be efficient emitters of PAHs and fine particle mass (Tissari et al. 2007 and 2009). There is a distinct weekly variation of the use of sauna stoves in Finland; these are most frequently used on Saturday afternoons and evenings. Also other fireplaces, except for boilers, are most frequently used during weekends. Clearly, more heating is needed during the colder period of the year, especially in winter; the rate of wood combustion is therefore much higher in colder periods. (Gröndahl et al., 2011; Kaski et al. 2016a)

Separate samples were therefore collected on Saturdays at a detached house area around the measurement site DH6 in 2013. On the average, the monthly averaged BaP concentrations were substantially higher on Saturdays, compared to the corresponding mean value during all days (Fig. 5). The values on Saturdays were clearly higher especially during part of winter months, in January and February, and part of summer, June and July. However, the values in June were based only on one sample on Saturday. These results indicate that the impacts of wood combustion are highly variable in time.

### 3.2 Predicted spatial concentration distributions

The spatial distributions of annual average BaP concentrations in the HMA originated from wood combustion were predicted for four years: 2008, 2011, 2013 and 2014. The results for two years, 2008 and 2011 are presented in Figs. 6a-b. The results for these two years were selected, as they represent the lowest and highest annual average concentrations due to the differences in meteorological conditions.

The influence of the regional and long-range transported background has not been allowed for in the values of these figures. In this way, the influence of the local emission sources can be visualized more clearly. The same emission values were applied for all years; however, the influence of the hourly variation of the meteorological factors on atmospheric dispersion was taken into account for all the target years. The predicted differences between the BaP concentrations are therefore due solely to the interannual variability in the meteorological conditions.

As expected, the spatial variation of the pollution distribution (Fig. 6) was closely associated with the corresponding variation of the emissions (Fig.1b), which in turn was closely associated with the density of detached and semidetached houses (Fig.





1a). The highest concentrations occurred in detached house areas, in which the highest annual average concentrations (in selected calculation grid with horizontal spacing of100 m x 100 m) were 1.0 and 1.3 ng m$^{-3}$ for 2008 and 2011, respectively. In 2011, the concentrations were clearly higher than in 2008. In the center of Helsinki, the annual average concentration from local wood combustion was below 0.2 ng m$^{-3}$.

Sanka et al. (2014) evaluated the spatial variation of the BaP concentrations using dispersion modeling and passive sampling in the city of Liberec in northern Czech Republic. However, the structure and spatial distribution of urban emissions in that city was substantially different compared with the one in the Helsinki Metropolitan Area. They also included local traffic and industrial emissions in the modelling. They found the highest predicted concentrations to occur in the city centre, and close to

10  a bitumen mixing plant.  They evaluated that the higher concentrations in the city centre were probably caused by both the higher density of local heating and traffic sources. According to their estimates, 95 % of the urban BaP emissions in winter were originated from domestic heating.

**3.3 Comparison of the observed and predicted annual average concentrations**

To compare the predicted and measured concentrations, a regional background concentration of 0.135 ng m$^{-3}$ was added to the computed concentrations from local residential combustion. This value is the median of the measured BaP concentrations at the regional background station of Hyytiälä (RB2) in southern Finland in 2009-2014. This regional background value was assumed to be a constant both in time and throughout the urban area.

The predicted and observed annual average concentrations are presented in Fig. 7. Only the days with measurements were taken into account. The predicted concentrations agreed fairly well with the measured concentrations for four detached house areas (DH2, DH3, DH6 and DH7). For two detached house areas (DH1 and DH5), the agreement was relatively worse. In case of the urban background site (UB), the agreement varied from year to year. For most stations and years (except for DH2 in

25  2008 and DH3 in 2011), the computed concentration was higher than the observed value.

One probable reason for the disagreements between modelled and observed concentrations is the inaccurate description of the temporal variation of emissions. The treatment of the temporal variation of emissions in the model is based on temporal variation coefficients (monthly, weekly and daily). However, the model does not take into account the influence of the daily

30  or annual variation of meteorological conditions (especially that of the ambient temperature) on the intensity of wood burning. In general, the uncertainty in emission factors can be substantial.  Such uncertainties are partly caused by the scarcity of experimental studies regarding the emissions of various fireplace types, especially for the stoves of saunas (Section 2.3). Partly, such uncertainties are also caused by the wide variation of emission factors in terms of the quality and processing of fuels, the quality and structure of heaters, and combustion techniques and procedures.





The model does not take into account the reactivity of BaP in the air. Heterogeneous reactions of BaP on the particle surfaces may have an effect on concentrations especially in summer (Keyte et al. 2013). However, the detailed chemical transformation equations of these reactions are still insufficiently known; it is also expected that most of the BaP molecules are located inside

the bulk particles and may therefore not be accessible for these reactions. Therefore BaP is expected to be removed mainly by dry and wet deposition of particles; the degradation of BaP through chemical reactions is expected to have a relatively smaller effect on the measured BaP concentrations.

Another factor which causes differences between modelled and observed concentrations is the spatial resolution of the

modelling; the emission sources were assumed to be located in grid squares with the size of 100m x 100m. However, measurements represent specific spatial points; especially in case of small-scale combustion, the measured values may be influenced by very local distributions of sources and other features.

The regional background concentration was based on the measured values at the station of Hyytiälä in southern Finland.

However, this site represents more continental conditions compared with the HMA, and it may not be sufficiently representative of the study domain.

**4  Conclusions**

Effect of local small-scale wood combustion on the BaP concentrations was studied, using ambient air measurements, emission estimates and dispersion modelling. Measurements were conducted at 12 different locations during the period from 2007 to 2015. A novel emission inventory was compiled for the small-scale wood combustion in the HMA in 2014 and the spatial

distributions of annual average benzo(a)pyrene concentrations originated from wood combustion were predicted for four meteorologically different years: 2008, 2011, 2013 and 2014.

Both the measurements and the dispersion modelling showed that the European Union target value for the annual average BaP concentrations (1 ng m$^{-3}$) was clearly exceeded in part of the detached house areas. The WHO reference level for BaP (0.12

ng m$^{-3}$) was exceeded at all urban and suburban sites every year and at rural background sites during most of the years. The predicted annual average concentrations agreed fairly well with the measured concentrations.

At street canyons, the measured concentrations of BaP were at the same level as urban background, being clearly lower than those in suburban detached house areas. This indicates that the influence of local vehicular traffic on the BaP concentrations





is very small, or almost negligible, in the street environments of the HMA. The measured BaP concentrations were also highly correlated with the measured levoglucosan concentrations supporting the finding that wood combustion is the dominant source. Regionally and long-range transported pollutants also have a notable impact on BaP concentrations in the HMA and southern Finland.

The concentrations of BaP were clearly higher on Saturdays, when the stoves of saunas and other combustion devices are frequently used. Saunas are very common in Finland; they therefore probably have a more pronounced impact on the concentrations of BaP than in any other country globally. The substantial influence of the stoves of saunas was one of the reasons why wood combustion emissions were found to be highly variable in time and space.

Based on both measurements and modeling, it can be concluded that wood combustion was the main local source of ambient air BaP in the HMA. Local wood combustion was found to have a substantially more important role for the concentrations of BaP, compared with those for $PM_{2.5}$.

Combining information obtained from ambient air measurements, wood combustion emission estimates and atmospheric dispersion modeling enabled a quantitative characterization of the influence of residential wood combustion. The results can be used in urban and environmental planning, regarding the impacts of small-scale combustion; the results have also significance in view of the environmental and climate change mitigation policies.

Although the predicted and measured annual concentrations agreed fairly well, there are several research needs on the BaP emission and dispersion modelling. In the future studies, it would be valuable to quantitatively measure the BaP emission factors for sauna stoves and for various other fireplace types in various operating conditions, to reduce the uncertainly of emission estimates. It would also be useful to construct an emission model that would take into account the impact of the actual meteorological conditions especially that of the ambient temperatures on wood combustion activities for different types of
houses and fireplaces. That kind of emission model in combination with dispersion modelling could potentially substantially improve the accuracy of the BaP concentration predictions, especially regarding their temporal variations.

**Acknowledgements**

This study has been a part of the research projects APTA (The Influence of Air Pollution, Pollen and Ambient Temperature on Asthma and Allergies in Changing Climate), MMEA (Measurement, Monitoring and Environmental Efficiency Assessment) and NordicWelfAir (Project #75007: Understanding the link between Air pollution and Distribution of related Health Impacts and Welfare in the Nordic countries). The funding from the European Commission, the Finnish Funding Agency for Innovation, the Academy of Finland and the Nordforsk Nordic Programme on Health and Welfare is gratefully




acknowledged. The research was also supported by the Academy research fellow project (Academy of Finland, project 275608). The original wood combustion activity survey in the HMA was prepared in co-operation by the Work Efficiency Institute, the Helsinki Region Environmental Services Authority and the Finnish Environment Institute. We also acknowledge Ph.D. Jarkko Tissari from the University of Eastern Finland as well as Ph.D. Niko Karvosenoja and Ph.D. Kaarle Kupiainen
from Finnish Environment Institute for their expertise and co-operation in determination of emission factors.

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





Table 1: Information on the measurement sites, together with the mean BaP, PM$_{2.5}$ and NOx concentrations.

| Classification of site | Code | Name of site | Sampling year(s) | BaP (ng m$^{-3}$) | PM$_{2.5}$ (µg m$^{-3}$) | NOx (µg m$^{-3}$) |
|---|---|---|---|---|---|---|
| Detached house area 1 | DH1 | Vartiokylä | 2009-2015 | 0.6 | 7.5 | 21 |
| Detached house area 2 | DH2 | Itä-Hakkila | 2008 | 1.1 | - | - |
| Detached house area 3 | DH3 | Päiväkumpu | 2011 | 1.2 | 10.4 | 21 |
| Detached house area 4 | DH4 | Kattilalaakso | 2012 | 0.6 | 8.2 | 18 |
| Detached house area 5 | DH5 | Kauniainen | 2013 | 0.4 | 7.1 | 13 |
| Detached house area 6 | DH6 | Tapanila | 2013 | 1.0 | 8.8 | 22 |
| Detached house area 7 | DH7 | Ruskeasanta | 2014 | 1.0 | 10.8 | 19 |
| Detached house area 8 | DH8 | Lintuvaara | 2015 | 0.9 | 7.1 | 14 |
| Street canyon 1 | SC1 | Unioninkatu | 2007 | 0.3 | - | 76 |
| Street canyon 2 | SC2 | Töölöntulli | 2010 | 0.3 | 13.0 | 166 |
| Street Canyon 3 | SC3 | Mäkelänkatu | 2015 | 0.2 | 8.0 | 108 |
| Urban background | UB | Kallio | 2007-2015 | 0.3 | 7.8 | 28 |
| Rural background 1 | RB1 | Virolahti | 2007-2015 | 0.2 | 6.1 | 5.1 |
| Rural background 2 | RB2 | Hyytiälä | 2009-2015 | 0.1 | - | 2.2 |
| Remote background | RE | Pallas | 2009-2015 | 0.03 | 3.7* | 1.0 |

*only for years 2011, 2012, 2014 and 2015



a)

b)

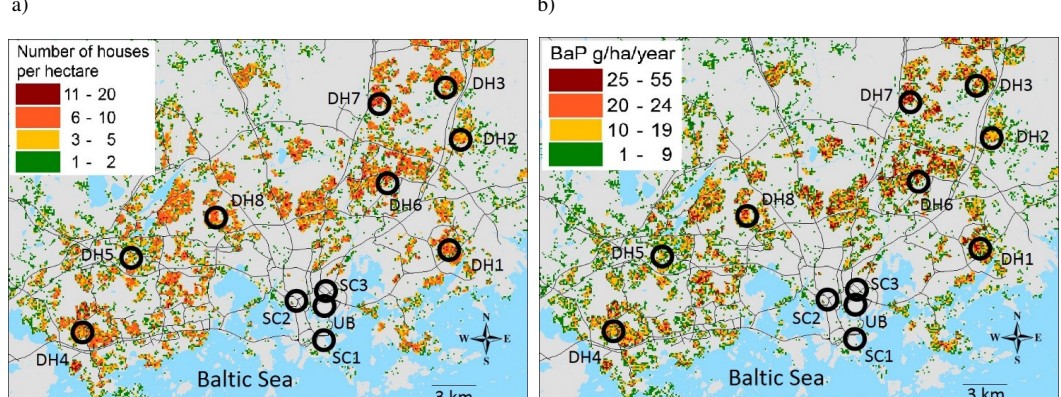

**Figure 1: Air quality monitoring sites (diameter of circles 1 km), a) the density of detached and semidetached houses and b) estimated yearly BaP emissions from wood combustion in the Helsinki metropolitan area in 2014. The main road network is shown for clarity.**



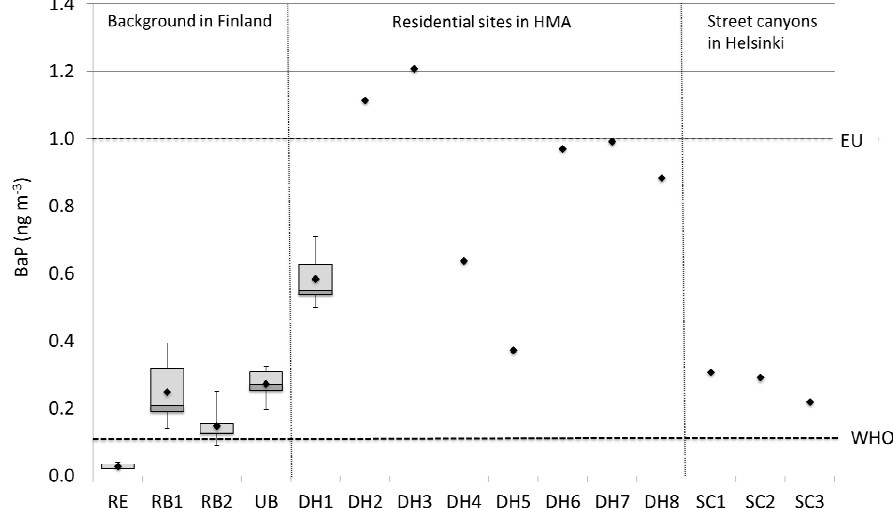

**Figure 2: Measured annual means of BaP concentrations (ng m⁻³) at different stations in Finland, compilation of results from 2007 to 2015. The box whisker plots represent the smallest value, the 0.25 percentile, the median value, the 0.75 percentile and the largest value for each measuring site. There was only one measured annual mean value at the sites, for which whisker plots have not been presented. The EU target value of 1 ng m⁻³ and the WHO reference level 0.12 ng m⁻³ are marked as dashed lines. (HMA=Helsinki metropolitan area)**





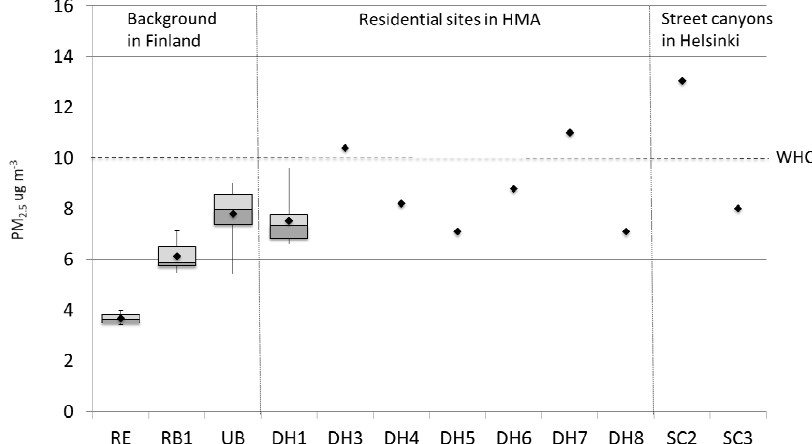

**Figure 3: Annual means of PM$_{2.5}$ concentrations (µg m$^{-3}$) at different BaP measurement stations in Finland. The box whisker plot represents the smallest value, the 0.25 percentile, the median value, the 0.75 percentile and the largest value for each measuring site. The WHO air quality guideline is marked as dashed line. The presented PM$_{2.5}$ data is for the same years as for BaP in Fig. 2 and Table 1, except for the value at the station RE, which is only for the years 2011, 2012, 2014 and 2015. (HMA=Helsinki metropolitan area)**





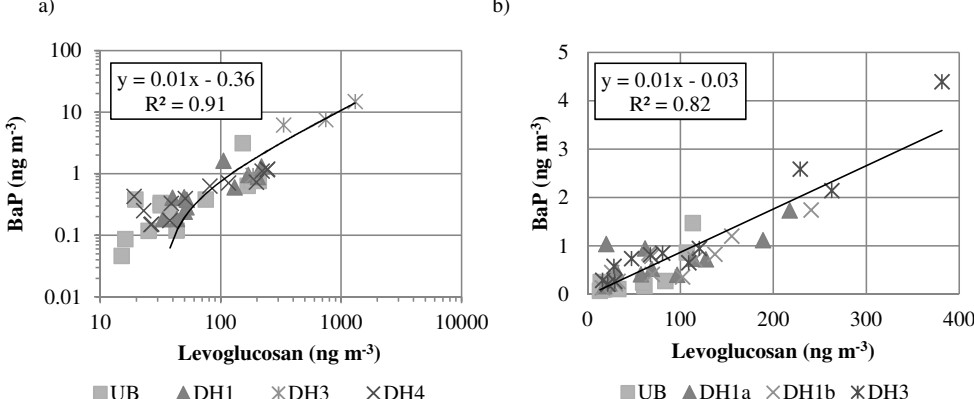

**Figure 4: The linear correlation between BaP and levoglucosan concentrations in the urban background (UB) and suburban detached house areas (DH) in Helsinki metropolitan area for a) 24 h means in February 2012 and for b) monthly means in 2011. Panels (a) and (b) have been presented on logarithmic and linear scales, respectively. The number of data points was 41 (panel a) and 48 (panel b).**



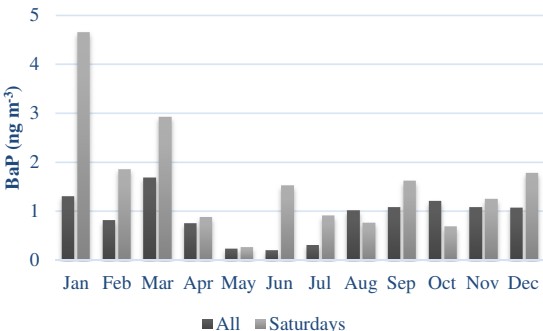

**Figure 5: Monthly mean BaP concentrations together with monthly mean concentrations on Saturdays at the detached house area around measurement site DH6 in Helsinki in 2013.**

30

35



(a)

(b)

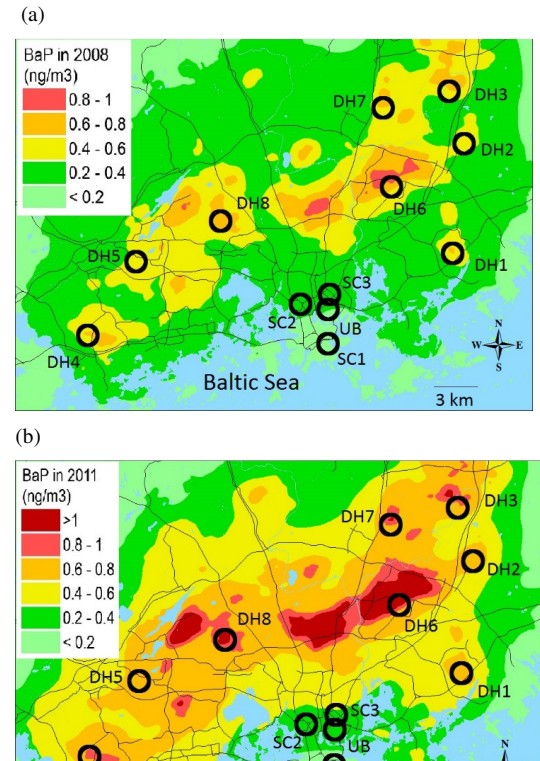

**Figure 6: The predicted spatial distributions of the annual average concentrations of BaP originated from wood combustion in the Helsinki metropolitan area in 2008 (a) and 2011 (b). The influence of the regional and long-range transported background is not included in the values of these figures. The main road network is shown for clarity.**





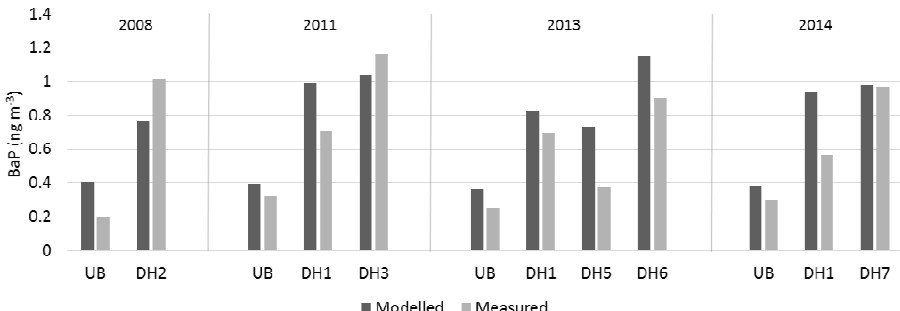

**Figure 7: Comparison of predicted and observed annual average BaP concentrations at different sites in Helsinki metropolitan area in 2008, 2011, 2013 and 2014.**

