# Peer review of "Evaluation of the impact of wood combustion on benzo(a)pyrene (BaP) concentrations; ambient measurements and dispersion modelling in Helsinki, Finland"

_Atmospheric Chemistry and Physics, 2016_

## Referee Comment (RC1) · Anonymous Referee #1 · 30 Nov 2016

Review of the manuscript entitled: "Evaluation of impact of wood combustion on benzo(a)pyrene concentrations, using ambient air measurements and dispersion modelling in Helsinki, Finland" by Hellén et al. report medium yearly and monthly concentrations of benzo(a)pyrene and uses these concentrations and inventory data from a questionnaire and emission inventory in a simple dispersion model in order to evaluate the influence of wood burning in sub-urban areas in the Helsinki Metropolitan Area (HMA).

General Comments. In a pre-review of the manuscript it was already mentioned that

many variables, such as emission strength of sources, meteorology and chemical properties of analyzed compound are not well addressed and could cause uncertainties and discrepancies among predicted and measured data. However, the strength of the study is situated in the relationship between the "wood-burning" marker; levoglucosan, and benzo(a)pyrene, and the selection of urban sites and sub-urban sites. Although the authors show the strong correlation between the compounds, it can not appropriately quantify the apportion to BaP form other sources.

An important part of the manuscript and results is based on a questionnaire and data from studies that have been published by the "Helsinki Region Environmental Services Authority" in Finnish and can not be consulted without knowledge of this language. It is not clear what the uncertainties are of the data, and this is also not included in the presented manuscript. Especially, when comparing the results of measured versus calculated values, it is important to mention these uncertainties in order to get an insight on the quality of the outcome. Nevertheless, the around 800 household that were asked on their use of wood combustion gave a clear idea about the importance of this activity. Moreover, there seems to be no doubt that wood burning is an important source for benzo(a)pyrene in the HMA, which on its term could be useful in the discussion of the influence of semi-urban / semi-rural areas on regional air quality, since biomass burning is promoted as an energy source in the European Union.

The applied model is suitable for the studied region, but could be given more discrepancies in areas which are exposed to multiple sources and where wood burning will not be so dominant.

It would have been interesting to show more monthly data of all D-sites as well as U-sites to get a clearer overview on the results and the relationship between benzo(a)pyrene and levoglucosan, especially in relationship with co-variables, such as meteorological data.

Besides these general comments, the following doubts should be clarified:

[Figure]

Specific comments. Introduction: Pg2.ln8. A reference is missing on the trends in PAH in others parts in EU. In fact Southern Sweden and Northern Finland are not the only regions/sites were PAH (or specific BaP) is not decreasing, and many of these areas face similar situations as in the present study; i.e. combustion of wood or / and coal. Comment this here.

Ln.28. The outcome of the predicted BaP concentrations for the studied area should be mentioned and discussed here. What did the model predict for the studied areas, and was this related to wood combustion?

Pg3. ln2. The studies mentioned here are based on BaP and levoglucosan (and other compounds) measurements, and indicate that there are areas in EU which have high apportion of wood combustion for PAH.

Ln.8 to 11. This part could be left out from the manuscript, since it deals with PM2.5 and not BaP.

Ln.11 to Ln15 "In the Helsinki. . .sauna stoves". This sentence should be removed since it deals with PM2.5.

Ln23. "The very few studies" dealing with the "quantitative effect of residential wood combustion on the ambient air concentrations of PAHs" should be mentioned here.

Ln24. What do the authors consider "reliable estimations of the spatial distribution and temporal variations"? How are these items addressed in the present study? In my view, the authors present many sampling sites, and many sampling years, but few sites are sampled every year. This result is a mix of data that may not improve the reliability of the outcome. Discuss this in the manuscript.

Ln27. The authors could rewrite the sentence to: "Wood combustion is a major source of PAH (Shen et al. 2013), although the emission rates depend heavily on a large variety of factors, such as . . ."

Pg4.Ln2. The "new inventory" should be discussed and compared to "old" ones.

Ln3. There is a reference missing for the levoglucosan analysis in ambient air PM. Moreover it is not clear to me why the data of black carbon was not used in the present study, since this measured in the emissions (Gröndahl et al. 2011) in considerable amount (Savolhathi, et al. 2016). The used model should be introduced here.

Methods: Ln17. There should be a reference or measurements that demonstrate the "minor impact on air quality".

Measurements methods: Pg5.ln14. What is the uncertainty of the analysis at 0.200 ng/m3, and what do the authors mean with "estimated" measurement uncertainty? Please, clarify.

Ln10. Why do you want to pool samples? This will reduce the information of the sample day. Why was this done?

Ln20. It is not clear from the text if the samples of PAH and levoglucosan were the same filter, or different filter samples. It is also not clear if the samples for levoglucosan and PAH were collected on the same day and site. This should be clarified here and the sampling strategy should be discussed.

Ln25. It is not clear to me why the authors do not want to evaluate the emission from traffic. They use arguments, but here they have the tools to quantify the contribution. Please, clarify, why you do not want to quantify this contribution.

Ln31 (but also other issues mentioned from on pg.6 to pg7 referring to Kaski 2016). The report should be explained here in more detail, since this inventory is the fundament of your results. The mentioned report for more details is in Finnish, and many people do not control this language. It is also mentioned here that the black carbon emissions were estimated. It would have of major interest to show the results of BC measurements in relation to BaP and other tracers for the apportion of wood combustion (and other sources) on the ambient air.

Pg6. Ln1. What is the influence of these factors on the results in this study? Explain in

more detail.

Ln31. What are the uncertainties of these factors in the present study? Is it possible to introduce them in the final result, so the reader understands the error of the calculations and will be able to validate the model better?

Pg7.ln12. What do you mean you did not get enough information form the questionnaire to estimate the influences of meteorological variables on the emissions? These variables have influence on emissions (also see pag.10.ln30 and pg11.ln27). Clarify the reliability the questionnaire.

Atmospheric dispersion modelling: Pg7.ln21. It is not clear why the model was run on an hourly base while BaP levels are monthly concentrations. Please, clarify.

Pg8.ln6. Why was particle bounded BaP treated as an inert gas?...why not as an inert particle, or a reactive particle? Discuss the differences between these possibilities and the influence on the outcome of the model.

Correlations of the concentrations of BaP and levoglucosan Pg9.ln23. Here it is mentioned that levoglucosan may not be a quantitative tracer due to its reactivity and dependence of combustion conditions, but this could be compared to BaP, which exhibit similar properties. Are they comparable?

Pg.10.ln1. The observed ratio in the present study should be compared to more than one study (Belis et al, 2011). In fact, the Belis study is also based on measurements, like the present study. The observed difference of a factor 10 should therefore be discussed in other terms. It is important here. For your interest; Fine et al. 2004. ENVIRONMENTAL ENGINEERING SCIENCE 21. observed BaP to levoglucosan ratios closer to 0.001 then 0.01. Why was the relationship between BaP and levoglucosan not used to "estimate" the contribution of wood combustion throughout the year and daily, as was done elsewhere (see Belis et al, 2011, or van Drooge & Perez Ballesta. 2009. ENVIRONMENTAL SCIENCE AND TECHNOLOGY.43.7310)?

[Figure]

Predicted spatial concentration distributions: Pg10.ln24 (and second paragraph). Is not clear how the "differences in meteorological conditions" influence the high and low BaP levels. What are these conditions and how are they different. Please, clarify.

Pg11.ln6 to 12. The comparison to the Czech study is almost irrelevant, since this is another situation, other sampling method and traffic included-model. The comparison could be removed from the manuscript. Are there no other studied to compare, and what would happen with the model outcome if traffic emissions were included?

Comparison of the observed and predicted average annual concentrations. It is not clear why only annual results are compared? Why not monthly results, or at higher temporal resolution. It is interesting to see how the BaP concentrations fluctuate along the year in the different months (or weeks, or days, such as weekend versus weekdays, in the HMA) It is unclear why 0.135 ng/m3 was add to the "computed concentration". If this is background, where does it come from? It is a considerable level. Why the regional background from Hyytiaäla was used and not a regional urban background from a urban background station, or urban site with low levels (as observed in this study)?

Ln.27 and whole paragraph. It is not clear why the temporal variation of the emissions were not addressed better. If this emissions are based on daily to monthly variations (not really clear how), it is not clear why this was not possible to investigate the influence of the meteorological conditions on the emissions.

The authors declare that many factors, such as meteorological influences on emissions, reactivity of BaP, particle-bounded properties of BaP, and the use of a regional background in the vicinity of the studied area were not taken in to consideration when they started the modelling, but these factors are well known beforehand. Can the authors improve their model? Really, mentioning these limitations in the last part of the discussion is not appropriate. These questions are clear from the beginning and should have been introduced right from the start, or the fact that they were not introduced (as

was the case here) they should have been mentioned and discussed beforehand. The model is probably not that bad, but misses a clear uncertainty calculation which makes the discussion too speculative.

I recommend to authors to focus on the real results, the chemical analysis that show that the sub-urban (detached house) zone is an area of seriously high BaP concentrations which is link directly to wood burning based on levoglucosan levels. This wood burning is confirmed by the habitants of the area which declare in the questionnaire that they use wood for (secondary heating) and saunas. These saunas are probably typical in Nordic countries, while in other areas domestic heating is more important. It does not matter. The results from the questionnaire are used in a simple model to see if this wood burning coincides with the measured BaP concentrations. They do, but it is not clear to what extend..since there is an error calculation missing.

The authors could mention that the applications of questionnaires in the case of wood burning are very powerful, since a national/regional inventory on wood burning is not existing or underestimating the real wood consumption, since the wood is often non-invoiced and self-supplying wood. In the context of the present study a comparison with other questionnaires could be made, such as the one by Pastorello et al. 2011. ATMOSPHERIC ENVIRONMENT. 45.2869–2876.

---

## Referee Comment (RC2) · Anonymous Referee #2 · 8 Dec 2016

The manuscript presents an interesting overview of measurements of PAHs and particles (PM2.5) in Helsinki Metropolitan area. The most notable about the study is that it covers eight years of measurements at several sites and the results are then combined with modelling to evaluate the levels over the whole area. As such the study is interesting and should be published. The largest problem with the manuscript is the inadequate treatment of uncertainty/variation in the presentation. Averages should be presented together with the uncertainty/variation related to the data, in both text and figures. Typically the standard deviation should be given. Furthermore the text should be checked by a native English-speaking person to correct grammatical errors. I have
noted a few here, but not all.

Specific comments:

Abstract: Explain all abbreviations the first time they occur.

Page 2 line 8-9: "This could be related to the ongoing and not decreasing sources." This is an unspecific and unclear statement. Please clarify.

P2 line 15: Change to: If we consider the reference level...

P2 line 20: Add relevant references to the first sentence. The first sentence of this paragraph is on PAH, while the rest of the paragraph presents the study of Butt et al. on PM, not PAHs. I suggest to gather the discussion of previous studies according to the components. It would also be useful to present information on emission inventories, especially for Finland.

P3 L9: Please consider the effect of improved stoves leading to lower emissions per kg wood combusted.

P3 L11: Unclear sentence which needs revision.

P3 L24-25: Please indicate here how this is overcome in the present work.

P4 site descriptions: It would be useful to provide more information on the distance between measurement sites in non-urban areas and nearby houses with wood stoves.

P5 Section 2.2: Please provide more information on the measurement methods, especially the analysis methods for both PAH and levoglucosan (columns, temperature program, standards and so on). Line18: Did you really use deionized water and not MilliQ water?

P6 L5: the amount of the wood combusted -> the amount of wood combusted...

P6 L19-22: Please provide the standard deviations for these averages.

P8 L19: have been -> are

P9 L5-6: Is this data shown?

P10 L1-4: Please clarify this paragraph.

P11 L19: Please discuss the uncertainty associated with this approach.

P11 L30: Is there previous studies that show this effect?

P11L31: In general -> Furthermore. Since this is a different problem you discuss.

P11 L34. Please cite relevant literature here.

P12 : Please discuss your results compared to other previous studies.

P13 L7-8: Please improve this sentence.

Table 1 and figures: Please provide standard deviations when possible. The points in Figures 2 and 3 are annual values and the variation should be clearly stated (even though it might not be feasible to make box and whisker plots). Bars in Fig. 5 also lacks indication of the associated variation.
* * *

---

## Author Comment (AC1) · 1 Feb 2017

Thank you for the very good comments. We have carefully considered all the comments and improved our manuscript accordingly. Please, find below our answers to the general and specific comments.

General Comments. In a pre-review of the manuscript it was already mentioned that many variables, such as emission strength of sources, meteorology and chemical properties of analyzed compound are not well addressed and could cause uncertainties and discrepancies among predicted and measured data.

-We have substantially improved the discussion of sources, meteorology and chemical properties in the revised manuscript, as detailed below.

However, the strength of the study is situated in the relationship between the "wood-burning" marker; levoglucosan, and benzo(a)pyrene, and the selection of urban sites and sub-urban sites. Although the authors show the strong correlation between the compounds, it can not appropriately quantify the apportion to BaP form other sources.

-We totally agree with the reviewer. The BaP vs. levoglucosan comparison was not done to apportion quantitatively the source contributions to BaP (and trying to do that would not be justified). This comparison only gives an indication (qualitatively) that the sources of these substances may be partly the same. We have revised the interpretation of these results to be more accurate and more cautious (section 3.1.2, two first paragraphs).

An important part of the manuscript and results is based on a questionnaire and data from studies that have been published by the "Helsinki Region Environmental Services Authority" in Finnish and can not be consulted without knowledge of this language.

-We would just like to comment that some of the authors are from this Authority, and have actually conducted this inventory themselves. This manuscript actually tries to report the key results of this inventory (and the referenced report) in English.

It is not clear what the uncertainties are of the data, and this is also not included in the presented manuscript. Especially, when comparing the results of measured versus calculated values, it is important to mention these uncertainties in order to get an insight on the quality of the outcome. Nevertheless, the around 800 household that were asked on their use of wood combustion gave a clear idea about the importance of this activity.

-Unfortunately, the original emission inventory of Kaski et al. (2016a) does not include a quantitative evaluation of the uncertainties. However, we have added a discussion

on these uncertainties based on previous relevant national studies on this topic. We have, e.g., added standard deviation values to the average amounts of wood burned (section 2.3, p.7).

Moreover, there seems to be no doubt that wood burning is an important source for benzo(a)pyrene in the HMA, which on its term could be useful in the discussion of the influence of semi-urban / semi-rural areas on regional air quality, since biomass burning is promoted as an energy source in the European Union.The applied model is suitable for the studied region, but could be given more discrepancies in areas which are exposed to multiple sources and where wood burning will not be so dominant.

-Yes, we agree: In some other areas, it could certainty prove to be necessary to allow for the influence of e.g. industrial sources or vehicular sources in much more detail.

It would have been interesting to show more monthly data of all D-sites as well as U-sites to get a clearer overview on the results and the relationship between benzo(a)pyrene and levoglucosan, especially in relationship with co-variables, such as meteorological data.

-Monthly data for all U and D sites and all years can be found as supplement Fig. S1 and S2.

-However, we did not perform any co-variable analyses. Unfortunately, the dataset is too small for drawing any statistically valid conclusions in that respect.

Specific comments.

Introduction: Pg2.ln8. A reference is missing on the trends in PAH in others parts in EU. In fact Southern Sweden and Northern Finland are not the only regions/sites were PAH (or specific BaP) is not decreasing, and many of these areas face similar situations as in the present study; i.e. combustion of wood or / and coal. Comment this here.

-We corrected this sentence to cover whole EU area and added a reference.

Ln.28. The outcome of the predicted BaP concentrations for the studied area should be mentioned and discussed here. What did the model predict for the studied areas, and was this related to wood combustion?

-it is now mentioned in the manuscript highest values (>0.4 ng/m3) were predicted for central and eastern Europe and that highest concentrations are mostly due to wood combustion.

Pg3. ln2. The studies mentioned here are based on BaP and levoglucosan (and other compounds) measurements, and indicate that there are areas in EU which have high apportion of wood combustion for PAH.

-this comment was added to the text

Ln.8 to 11. This part could be left out from the manuscript, since it deals with PM2.5 and not BaP.

-This part was removed

Ln.11 to Ln15 "In the Helsinki. . .sauna stoves". This sentence should be removed since it deals with PM2.5.

-This part was removed

Ln23. "The very few studies" dealing with the "quantitative effect of residential wood combustion on the ambient air concentrations of PAHs" should be mentioned here.

-this was added to the text

Ln24. What do the authors consider "reliable estimations of the spatial distribution and temporal variations"? How are these items addressed in the present study? In my view, the authors present many sampling sites, and many sampling years, but few sites are sampled every year. This result is a mix of data that may not improve the reliability of the outcome. Discuss this in the manuscript.

-The first mentioned sentence was somewhat unclear. We have revised it, to refer specifically to modelling.

-We also revised our reference to the present study to be more cautious (only to say what we attempt to do): "In the present study, we attempt to combine several years of measurement data from different stations with dispersion modelling to overcome this issue."

-We have presented data for eight years for several sites, which is a substantially extensive data set if we consider specifically the measurements of BaP. Due to laborious and expensive measurements of BaP, there are no extensive data sets (i.e. longer term that in this study) available for BaP.

-It is correct (as the reviewer states) that the dataset is not perfect in terms of homogeneity. This is difficult to achieve in practice in a rapidly evolving urban area, and also due to the resource limitations. However, the fact that not all sites measured each year, has been taken into account in the analysis of the results, of course.

Ln27. The authors could rewrite the sentence to: "Wood combustion is a major source of PAH (Shen et al. 2013), although the emission rates depend heavily on a large variety of factors, such as . . ."

-the sentence was rewritten

Pg4.Ln2. The "new inventory" should be discussed and compared to "old" ones.

-We have revised this sentence to clarify the importance of the new inventory. The word 'new' was changed to 'novel'.

Ln3. There is a reference missing for the levoglucosan analysis in ambient air PM. Moreover it is not clear to me why the data of black carbon was not used in the present study, since this measured in the emissions (Gröndahl et al. 2011) in considerable amount (Savolhathi, et al. 2016). The used model should be introduced here.

-A reference on levoglucosan was added.

-The used model was introduced.

-Black carbon data for ambient air was not available for most of the sites. For instance, there are only two annual BC data sets available for detached house areas in the Helsinki metropolitan area; Vartiokylä in year 2009 and Ruskeasanta in year 2014. An analysis based on this dataset would have therefore not yielded reliable conclusions.

Methods: Ln17. There should be a reference or measurements that demonstrate the "minor impact on air quality".

-A suitable reference for source contributions was added (i.e. Soares et al., 2014)

Measurements methods: Pg5.ln14. What is the uncertainty of the analysis at 0.200 ng/m3, and what do the authors mean with "estimated" measurement uncertainty? Please, clarify.

-uncertainty calculation was clarified and reference to standard EN15549 (2008), which was followed, was added. Uncertainty at the concentration level 0.2 ng/m3 was added.

Ln10. Why do you want to pool samples? This will reduce the information of the sample day. Why was this done?

-Samples were pooled to save resources. Sample preparation and analysis of these samples is laborious and expensive. If the main aim is to get annual mean concentrations, pooling to monthly samples is feasible.

Ln20. It is not clear from the text if the samples of PAH and levoglucosan were the same filter, or different filter samples. It is also not clear if the samples for levoglucosan and PAH were collected on the same day and site. This should be clarified here and the sampling strategy should be discussed.

-We added that samples were collected on same days and at same sites as BaP samples

Ln25. It is not clear to me why the authors do not want to evaluate the emission from traffic. They use arguments, but here they have the tools to quantify the contribution. Please, clarify, why you do not want to quantify this contribution.

-We have actually modelled the BaP concentrations from traffic in this area (within the EU funded TRANSPHORM study). However, the concentrations were negligible both compared with the contributions from residential combustion and regional background. The emission coefficients for vehicular traffic are also substantially uncertain. However, we revised this point by adding a reference to these computations.

Ln31 (but also other issues mentioned from on pg.6 to pg7 referring to Kaski 2016). The report should be explained here in more detail, since this inventory is the fundament of your results. The mentioned report for more details is in Finnish, and many people do not control this language. It is also mentioned here that the black carbon emissions were estimated. It would have of major interest to show the results of BC measurements in relation to BaP and other tracers for the apportion of wood combustion (and other sources) on the ambient air.

-We added more detailed information and discussion on the inventory to section 2.3.

-As it was already mentioned before, black carbon data for ambient air was not available for most of the sites. Therefore, it was not included in this study. However, we are currently studying the concentrations and sources of BC at two different sites in the Helsinki metropolitan area. We are planning to publish these new results within 1-2 years.

Pg6. Ln1. What is the influence of these factors on the results in this study? Explain in more detail.

-Discussion on effects of these factors and how we took them into account in this study, was added.

Ln31. What are the uncertainties of these factors in the present study? Is it possible to

introduce them in the final result, so the reader understands the error of the calculations and will be able to validate the model better?

-We added discussion on the uncertainties. Unfortunately, only a few studies provide BaP emission factors for typical fireplace types used in Finland, and these do not report sufficiently the inaccuracies of the measured values. Therefore, we considered that it was unfortunately not possible to present quantitative uncertainty estimates for the BaP emissions.

-As we already mention in the paper, it would be very important to perform new combustion experiment studies to achieve more robust knowledge on BaP emission factors and their inaccuracies, especially for sauna stoves and various other fireplace types, as well as for different burning conditions.

Pg7.ln12. What do you mean you did not get enough information form the questionnaire to estimate the influences of meteorological variables on the emissions? These variables have influence on emissions (also see pag.10.ln30 and pg11.ln27). Clarify the reliability the questionnaire.

-We have revised this paragraph. There was not sufficient quantitative information in the inventory regarding how peoples' heating habits change in terms of e.g. ambient temperature. It is clear that they heat more in cold weather, but how exactly this could be quantified for modelling is less clear.

Atmospheric dispersion modelling: Pg7.ln21. It is not clear why the model was run on an hourly base while BaP levels are monthly concentrations. Please, clarify.

-It is better to execute the model for each hour separately, to obtain a predicted time series of concentrations on a higher temporal resolution. The modelled results are not used only for a comparison with the monthly concentrations, but also for other purposes. The predicted averaged monthly concentration values are also more accurate, when these are based on the use of hourly meteorological data, instead of some more

coarsely averaged meteorology.

Pg8.ln6. Why was particle bounded BaP treated as an inert gas?...why not as an inert particle, or a reactive particle? Discuss the differences between these possibilities and the influence on the outcome of the model.

-The text has been revised to be more clear, as suggested. The term 'inert gas' is confusing, and it has been replaced with 'inert substance' in the revised text. In the dispersion modelling context, 'inert substance' means simply that the substance (which can be a particle or a gas molecule) transports and diffuses according to atmospheric turbulence, and no transformation processes are taken into account. On an urban scale, this is a reasonable assumption for BaP – but not necessarily on regional or large scales.

Correlations of the concentrations of BaP and levoglucosan Pg9.ln23. Here it is mentioned that levoglucosan may not be a quantitative tracer due to its reactivity and dependence of combustion conditions, but this could be compared to BaP, which exhibit similar properties. Are they comparable?

-Levoglucosan has been used as a tracer in many studies. However, the main problem in using it quantitatively in this study was much higher contribution of BaP in the sauna stove emissions than in other fireplaces. Saunas are commonly used in Finland, but we were lacking the information of levoglucosan emissions from sauna stoves. Discussion on this was added to the manuscript to section 3.1.2, last paragraph.

Pg.10.ln1. The observed ratio in the present study should be compared to more than one study (Belis et al, 2011). In fact, the Belis study is also based on measurements, like the present study. The observed difference of a factor 10 should therefore be discussed in other terms. It is important here. For your interest; Fine et al. 2004. ENVIRONMENTAL ENGINEERING SCIENCE 21. observed BaP to levoglucosan ratios closer to 0.001 then 0.01. Why was the relationship between BaP and levoglucosan not used to "estimate" the contribution of wood combustion throughout the year and

daily, as was done elsewhere (see Belis et al, 2011, or van Drooge & Perez Ballesta. 2009. ENVIRONMENTAL SCIENCE AND TECHNOLOGY.43.7310)?

-In Belis et al. (2011) ratio, which we mention in our manuscript, was average calculated from 10 different biomass burning studies ((Schauer et al., 2001; Fine et al., 2001, 2002, 2004; Hays et al., 2002, 2005; Dhammapala et al., 2007; Schmidl et al., 2008; Bari et al., 2009). It is not a ratio in ambient air, but average in the wood combustion emissions. This is now clarified in the manuscript (section 3.1.2, last para). We also added discussion on emissions of sauna stoves, where abundance of BaP have been found to be much higher than in many other appliance (Tissari et al. 2007).

- In our opinion, it is not possible to estimate the contribution of wood quantitatively using the relationship between BaP and levoglucosan, due to the dependence of the ratio on combustion conditions. As now also mentioned in the manuscript, the abundance of BaP in sauna stove emissions is much higher than in many other appliances, but since there are no ratio of levoglucosan and BaP in sauna emissions available, we cannot use this method for quantitative estimations. Saunas are commonly used in Finland and therefore they are expected to have substantial impact.

-Bari, Md. A., Baumbach, G., Kuch, B., Scheffknecht, G., 2009.Wood smoke as a source of particle-phase organic compounds in residential areas. Atmospheric Environment, 43, 4722e4732.

-Dhammapala, R., Claiborn, C., Jimenez, J., Corkill, J., Gullett, N., Simpson, C., Paulsen, M., 2007. Emission factors of PAHs, methoxyphenols, levoglucosan, elemental carbon and organic carbon from simulated wheat and Kentucky bluegrass stubble burns. Atmospheric Environment 41, 2660e2669

-Fine, P.M., Cass, G.R., Simoneit, B.R.T., 2001. Chemical characterization of fine particle emissions from fireplace combustion of woods grown in the north eastern United States. Environmental Science and Technology 35, 2665e2675.

-Fine, P.M., Cass, G.R., Simoneit, B.R.T., 2002. Chemical characterization of fine particle emissions from the fireplace combustion of woods grown in the southern United States. Environmental Science and Technology 36, 1442e1451.

-Fine, P.M., Cass, G.R., Simoneit, B.R.T., 2004. Chemical characterization of fine particle emissions from the wood stove combustion of prevalent United States tree species. Environmental Engineering Science 21, 705e721

-Hays, M.D., Geron, C.D., Linna, K.J., Smith, N.D., Schauer, J.J., 2002. Speciation of gas phase and fine particle emissions from burning of foliar fuels. Environmental Science and Technology 36, 2281e2295.

-Hays, M.D., Fine, P.B., Geron, C.D., Kleeman, M.J., Gullett, B.K., 2005. Open burning of agricultural biomass: physical and chemical properties of particle-phase emissions. Atmospheric Environment 39, 6747e6764.

-Schauer, J.J., Kleeman, M.J., Cass, G.R., Simoneit, B.R.T., 2001. Measurement of emissions from air pollution sources. 3. C-1eC-29 organic compounds from fireplace combustion of wood. Environmental Science and Technology 35, 1716e1728.

-Schmidl, C., Bauer, H., Dattler, A., Hitzenberger, R., Weissenboeck, G., Marr, I.L., Puxbaum, H., 2008. Chemical characterisation of particle emissions from burning leaves. Atmospheric Environment 42, 9070e9079

Predicted spatial concentration distributions: Pg10.ln24 (and second paragraph). Is not clear how the "differences in meteorological conditions" influence the high and low BaP levels. What are these conditions and how are they different. Please, clarify.

-We used the actual (hourly) met conditions during each year in the dispersion computations. These affect the results, e.g., during colder winters, there are commonly more stable and extremely stable conditions, and the concentrations are therefore relatively higher.

Pg11.ln6 to 12. The comparison to the Czech study is almost irrelevant, since this is

another situation, other sampling method and traffic included-model. The comparison could be removed from the manuscript. Are there no other studied to compare, and what would happen with the model outcome if traffic emissions were included?

-The Czech study is not directly comparable to the present study, and we therefore removed this para. We could not find any directly comparable studies.

Comparison of the observed and predicted average annual concentrations. It is not clear why only annual results are compared? Why not monthly results, or at higher temporal resolution. It is interesting to see how the BaP concentrations fluctuate along the year in the different months (or weeks, or days, such as weekend versus weekdays, in the HMA)

-We did not compare monthly averages, because we did not have monthly regional background estimates for the whole period starting from 2008. The measurements in Hyytiälä started in 2009, and therefore we used the median of measurements in 2009-2014 to estimate the annual average background. However, we did not consider this method possible for monthly averages, which may vary considerably from year to year.

It is unclear why 0.135 ng/m3 was add to the "computed concentration". If this is background, where does it come from? It is a considerable level. Why the regional background from Hyytiaäla was used and not a regional urban background from a urban background station, or urban site with low levels (as observed in this study)?

-This value is regional background. It needs to represent a value outside the urban area. Using a value inside the urban area would results in double-counting of the effect of urban emissions.

Ln.27 and whole paragraph. It is not clear why the temporal variation of the emissions were not addressed better. If this emissions are based on daily to monthly variations (not really clear how), it is not clear why this was not possible to investigate the influence of the meteorological conditions on the emissions.

-The influence of the meteorological conditions on the emissions depends on human behavior, which is unfortunately not known with sufficient accuracy.

The authors declare that many factors, such as meteorological influences on emissions, reactivity of BaP, particle-bounded properties of BaP, and the use of a regional background in the vicinity of the studied area were not taken in to consideration when they started the modelling, but these factors are well known beforehand.

-We have included a comment on the reactivity of BaP in section 2.4, last para. On an urban scale, this has a negligible effect on the concentrations. The particulate properties: if we assume that BaP is within fine PM (PM2.5), which is a reasonable assumption, the dry deposition can be neglected on an urban scale. Regional background was taken into account, using the best available measured data.

Can the authors improve their model?

-Yes. The influence of meteorological factors on the emissions (not only in the atmospheric dispersion as here) could be modelled in more detail (not only using fixed temporal profiles) in the next stage. However, this is not an easy task, and will require substantial research.

Really, mentioning these limitations in the last part of the discussion is not appropriate. These questions are clear from the beginning and should have been introduced right from the start, or the fact that they were not introduced (as was the case here) they should have been mentioned and discussed beforehand.

-These uncertainties have been mentioned and discussed in the methods section in the revised manuscript, such as section 2.4. The methods section is in our view the right place to mention the modelling uncertainties for the first time.

The model is probably not that bad, but misses a clear uncertainty calculation which makes the discussion too speculative.

-We agree that it would be useful to present a detailed uncertainty analysis of the modelling. However, some of the required input information is unfortunately not available. In particular, the uncertainties of the emission factors are not known sufficiently. This is caused by the difficulty of controlling the numerous factors that affect the emissions of small-scale combustion. This study only used the results of the emission studies, but it was not possible here to re-do emission uncertainty investigations.

I recommend to authors to focus on the real results, the chemical analysis that show that the sub-urban (detached house) zone is an area of seriously high BaP concentrations which is link directly to wood burning based on levoglucosan levels.

-The use of the levoglucosan (LG) measurements also have serious limitations. First, the ratio of BaP and LG depends on the sources, and the ratio is not even known for all source groups. This ratio also depends on the actual fuel used and the procedures of combustion. It is therefore not possible to solve the source apportionment issue by using the levoglucosan measurements. In brief, the LG measurements and useful, but can only be used as a supplementary method to understand the contributions from small scale combustion.

This wood burning is confirmed by the habitants of the area which declare in the questionnaire that they use wood for (secondary heating) and saunas. These saunas are probably typical in Nordic countries, while in other areas domestic heating is more important.

-Saunas are actually clearly more important in Finland, compared to the other Nordic countries.

It does not matter. The results from the questionnaire are used in a simple model to see if this wood burning coincides with the measured BaP concentrations. They do, but it is not clear to what extend..since there is an error calculation missing.

-We have studied the available national and international data and publications on this issue, and included the available information on the uncertainties (please see our

responses above). However, the state of the art does not make it possible to make a complete uncertainty analysis. This is a new research area, e.g., compared with the study of vehicular or industrial pollution.

The authors could mention that the applications of questionnaires in the case of wood burning are very powerful, since a national/regional inventory on wood burning is not existing or underestimating the real wood consumption, since the wood is often non-invoiced and self-supplying wood. In the context of the present study a comparison with other questionnaires could be made, such as the one by Pastorello et al. 2011. ATMOSPHERIC ENVIRONMENT. 45.2869–2876.

-as suggested by the reviewer comment on usefulness of questionnaires was added in the conclusions of the manuscript. -reference to Pastorello et al. 2011 was added into section 2.3.

---

## Author Comment (AC2) · 1 Feb 2017

Thank you for the very good comments. We have carefully considered all the comments and improved our manuscript according to them. Please, find below our answers to the general and specific comments:

The largest problem with the manuscript is the inadequate treatment of uncertainty/variation in the presentation. Averages should be presented together with the uncertainty/variation related to the data, in both text and figures. Typically the standard deviation should be given.

[Figure]

-We have added standard deviations and uncertainties whenever possible (please see also our answers to referee no 1 and for the specific comments below).

Furthermore the text should be checked by a native English-speaking person to correct grammatical errors. I have noted a few here, but not all.

-Native English-speaking person did now the language editing for the manuscript

Specific comments:

Abstract: Explain all abbreviations the first time they occur.

-we did this

Page 2 line 8-9: "This could be related to the ongoing and not decreasing sources." This is an unspecific and unclear statement. Please clarify.

-this unclear sentence was deleted

P2 line 15: Change to: If we consider the reference level... ÂÍ -We changed this

P2 line 20: Add relevant references to the first sentence. The first sentence of this paragraph is on PAH, while the rest of the paragraph presents the study of Butt et al. on PM, not PAHs. I suggest to gather the discussion of previous studies according to the components. It would also be useful to present information on emission inventories, especially for Finland.

-this paragraph was changed. The first sentence was combined with discussion on previous studies and part on PM was removed. There are no emission inventories for BaP and wood combustion available for Finland.

P3 L9: Please consider the effect of improved stoves leading to lower emissions per kg wood combusted.

-based on the recommendations other referee and our reconsideration, this paragraph was removed since it was discussing PM and not BaP.

P3 L11: Unclear sentence which needs revision.

-based on the recommendations other referee and our reconsideration, this paragraph was removed since it was discussing PM and not BaP.

P3 L24-25: Please indicate here how this is overcome in the present work.

-we added a sentence 'In this study we combined all ambient air data from 8 years of measurements with a recent emission inventory and dispersion modelling to better characterize wood combustion as a BaP source.'

P4 site descriptions: It would be useful to provide more information on the distance between measurement sites in non-urban areas and nearby houses with wood stoves.

-distance between measurement sites is shown in Fig. 1.

-better description of suburban measurement sites was added. In these 8 areas, the measurement sites were surrounded by detached houses. Wood combustion appliances are used in 90% of the detached houses in the HMA. Fig. 1 shows the number of houses per hectare. The nearest detached houses were located about 10-25 m distance from the measurement sites.

P5 Section 2.2: Please provide more information on the measurement methods, especially the analysis methods for both PAH and levoglucosan (columns, temperature program, standards and so on).

-better descriptions on measurement methods were added

Line18: Did you really use deionized water and not MilliQ water?

- word 'deionized' water was changed to 'MilliQ' water.

P6 L5: the amount of the wood combusted -> the amount of wood combusted...

-corrected

P6 L19-22: Please provide the standard deviations for these averages.

-we added standard deviations

P8 L19: have been -> are

-corrected

P9 L5-6: Is this data shown?

-This is shown by the box whisker plots in Fig. 2. Reference to this figure was added into the text

P10 L1-4: Please clarify this paragraph.

-more discussion on ratio between BaP and levoglucosan was added

P11 L19: Please discuss the uncertainty associated with this approach.

-Discussion on possible variation of background value has been added into text

P11 L30: Is there previous studies that show this effect?

-we were not able to find any previous studies showing this effect.

P11L31: In general -> Furthermore. Since this is a different problem you discuss.

-this was corrected

P11 L34. Please cite relevant literature here.

-references were added

P12 : Please discuss your results compared to other previous studies.

-discussion on concentration levels were added in the section 3.1.1.

P13 L7-8: Please improve this sentence.

-this sentence was changed

Table 1 and figures: Please provide standard deviations when possible. The points

in Figures 2 and 3 are annual values and the variation should be clearly stated (even though it might not be feasible to make box and whisker plots). Bars in Fig. 5 also lacks indication of the associated variation.

-we added standard deviations between the years into the table 1 and error bars showing the measurement uncertainty into figure 5.

-Seasonal variation is much higher than inter-annual variation. Therefore we think that showing the standard deviation calculated from the monthly variations for single points in figure 2 and 3 would make them unclear. In these figures annual values are compared with each other and with reference values and therefore we wish to keep box and whisker plots describing the inter-annual variation. Seasonal variations are shown in figure 5 and in supplement figures S1 and S2.
* * *

---

## Referee Report (RR1)

Review of acp-2016-780 Evaluation of the impact of wood combustion on benzo(a)pyrene (BaP) concentrations; ambient measurements and dispersion modelling in Helsinki, Finland by Heidi Hellén, et al. has been revised on the issues that were addressed in the first review.

In its present form the paper is suitable for publication in Atmospheric Chemistry and Physics.